# Monoethanolamine (MEA) Degradation: Influence on the Electrodialysis Treatment of MEA-Absorbent

**DOI:** 10.3390/membranes13050491

**Published:** 2023-05-01

**Authors:** Eduard G. Novitsky, Evgenia A. Grushevenko, Ilya L. Borisov, Tatiana S. Anokhina, Stepan D. Bazhenov

**Affiliations:** A.V. Topchiev Institute of Petrochemical Synthesis, Russian Academy of Sciences, Leninsky Prospect 29, 119991 Moscow, Russia

**Keywords:** alkanolamine, CO_2_ absorbent, electrodialysis, monoethanolamine, thermal degradation, membrane regeneration, fouling

## Abstract

The thermal-oxidative degradation of aqueous solutions of carbonized monoethanolamine (MEA, 30% wt., 0.25 mol MEA/mol CO_2_) was studied for 336 h at 120 °C. Based on the change in the color of the solution and the formation of a precipitate, the occurrence of thermal-oxidative degradation of the MEA solution with the formation of destruction products, including insoluble ones, was confirmed. The electrokinetic activity of the resulting degradation products, including insoluble ones, was studied during the electrodialysis purification of an aged MEA solution. To understand the influence of degradation products on the ion-exchange membrane properties, a package of samples of MK-40 and MA-41 ion-exchange membranes was exposed to a degraded MEA solution for 6 months. A comparison of the efficiency of the electrodialysis treatment of a model absorption solution of MEA before and after long-time contact with degraded MEA showed that the depth of desalination was reduced by 34%, while the magnitude of the current in the ED apparatus was reduced by 25%. For the first time, the regeneration of ion-exchange membranes from MEA degradation products was carried out, which made it possible to restore the depth of desalting in the ED process by 90%.

## 1. Introduction

The most common method for purifying gas mixtures from acidic impurities (mainly carbon dioxide and hydrogen sulfide) is their absorption with aqueous solutions of alkanolamines, primarily with monoethanolamine (MEA) solutions [1]. One of the disadvantages of this technology is the fact that MEA is subject to thermal and thermal-oxidative degradation at the desorption stage, which, as a rule, is carried out in the range of rather high temperatures of 110–150 °C. The specified desorption temperature range ensures the regeneration of amines to values that retain their effectiveness as absorbents. However, the use of elevated temperatures, on the one hand, ensures the depth of regeneration of amine solutions, and, on the other hand, provokes their thermal-oxidative destruction with the formation of ionic products and various oligomeric-type compaction products. The results of systematic studies of these processes carried out by G.T. Rochelle et al. are presented in a series of publications [2,3,4,5]. The schemes below illustrate the degradation processes and the resulting products of the interaction between MEA and CO_2_ [3].

In the absorber, MEA reacts with CO_2_ to form carbamate according to Figure 1 (the reverse reaction takes place in the desorber).

However, in some cases, MEA-carbamate by the cyclization reaction (also reversible) forms oxazalidone (Figure 2).

On the other side, MEA-carbamate can react with MEA to form N,N′-di(hydroxyethyl)urea (Figure 3).

In turn, 2-hydroxyzolidone can also react with MEA to form 1-(2-hydroxyethyl)-2-imidazolidone (HEIA) according to the reaction (Figure 4).

HEIA, as a result of the hydrolysis reaction, can form N-(2-hydroxy)-ethylenediamine (HEEDA) (Figure 5).

These four substances (2-oxazalidon, dihydroxyethylurea, HEIA, and HEEDA) and their further degradation products, including MEA dimers, trimers, and tetramers, as well as their cyclization, polymerization, and oxidation products, constitute the main group of degradation products [6,7]. The estimation of the scale of MEA losses in the processes of the absorption–desorption cycle due to the processes of thermal and thermo-oxidative degradation of MEA remains relevant [8,9,10]. Carboxylic acids (formic, acetic, oxalic, etc.)—the main component of the products of thermo-oxidative degradation, reacting with MEA to form heat stable salts (HSS)—are stable compounds that do not decompose under typical conditions of the desorption process and constantly accumulate in a closed system of absorption cycles. It is noted that an increase in the desorption temperature from 100 °C to 150 °C leads to an increase in the concentration of 1-(2-hydroxyethyl)-2-imidozolidone in the degradation products. The authors also note significant losses of MEA as a result of degradation with increasing temperature; for example, at 100 °C, MEA losses are within 2%, and at 150 °C, they reach 90%. In addition to the anions of carboxylic acids, the composition of HSS includes anions of inorganic acids (hydrochloric, nitric, sulfuric, and thiosulfuric). Such acids enter the amine, for example, with flue gas [11].

The contribution of HSS to the total losses of MEA during the absorption–desorption cycle can reach 80 g of a 30% MEA solution per ton of absorbed CO_2_ [3]. This means that when cleaning flue gases, for example, a 250 MW coal-fired thermal power plant (CO_2_ emissions of 49 kg/s), the loss of absorbent (30% MEA solution) will be 120 tons per year of operation, which is about 10% from the initial content of MEA in solution. As a result, technologies aimed at removing degradation products of alkanolamines are relevant. The main technologies are ion exchange, thermal distillation, and electrodialysis [12,13,14]. Membrane processes as environmentally friendly techniques have attracted much attention in recent years. The results of systematic studies [15,16,17,18,19,20] allow us to assume with cautious optimism the practical validity of the practical application of electrodialysis for the regeneration of MEA solutions.

In recent years, researchers have been interested in approaches to optimizing the process of HSS electrodialysis recovery from the point of view of reducing alkanolamine losses and reducing energy costs. Thus, a two-stage scheme for the electrodialysis isolation of TSS was proposed in order to reduce the losses of the amine. This approach makes it possible to reduce MEA losses by 30% [16]. In the work of the authors of [19], it was shown that during the regeneration of 30% MEA, the lowest energy consumption can be achieved at an initial degree of carbonization of the solution of 0.1 mol (CO_2_)/mol (MEA) −25.9 MJ/kg (absorbent), taking into account the energy consumption for additional desorption solution. The efficiency of the HSS isolation process using electrodialysis can be increased by adding an equimolar alkali. For the first time, the scheme with the introduction of a neutralization unit was proposed by the ElectroSep company, which currently presents it on the market for alkanolamine purification technologies under the ElectroSep^®^ brand. In [21], devoted to this process, both the increased complexity of the circuit and a significant reduction in the loss of alkanolamine, due to the supply of alkali directly into the membrane package and the absence of its mixing with the stream to be purified, are noted. Pilot tests [22] also showed an increase in the efficiency of HSS removal from MDEA solutions of 30% with the addition of NaOH. Another approach to reducing the off-target transfer of alkanolamine across the membrane is to fill the desalting chamber with a cation-exchange resin. Thus, in the work [23], it was shown that filling the desalting chamber with an ion-exchange resin made it possible to increase the efficiency of HSS recovery by 28.57% compared to the classical configuration of the electrodialyzer and by 7.88% compared to the configuration combined with neutralization. Moreover, MDEA losses decreased from 21.07% for the classic scheme to 3.78% with ion-exchange resin. The articles [24,25] propose the use of bipolar electrodialysis to reduce alkanolamine losses. The use of bipolar electrodialysis makes it possible to increase the current density during the transfer of HSS components in comparison with classical electrodialysis. This effect is due to the fact that water dissociation occurs on the bipolar membrane, so the alkanolamine is not transferred directly to the concentration chambers [26,27]. However, this does not allow one to expect a zero concentration of alkanolamine in the concentration chamber. As was shown in the work [28], there is a mechanism of ammonium ion transfer through an anion-exchange membrane under certain boundary conditions.

Another problem characteristic of the electromembrane desalination technology is the degradation of the surface material of ion-exchange membranes. One of the main reasons for the electrochemical degradation of an anion-exchange polymer is the nucleophilic attack of quaternary amines by hydroxyl ions [29], which are always present in aqueous solutions due to water dissociation. The result of such reactions is the transformation of a part of the quaternary amino groups into secondary and tertiary amines, as well as the elimination of fixed groups from the polymer matrix, breaking the carbon chains of the polymer matrix. The result of these reactions is the partial or complete destruction of the ion-exchange polymer. However, such behavior was not observed for MA-41 anion-exchange membranes (Shchekinoazot, Pervomajskij, Russia). The authors of the work [30] associated this with the regular crosslinking of the polymer ion-exchange material. However, in the case of absorption solutions of alkanolamines, precipitation is observed on the surface of ion-exchange membranes during electrodialysis [31,32,33,34]. Precipitation on the membrane leads to a decrease in current efficiency. The researchers note that both anion- and cation-exchange membranes are clogged. Both amine compaction products and insoluble heavy metal compounds settle on the membrane surface.

An analysis of the literature data on these processes indicates that the separation capacity of ion-exchange membranes (productivity and selectivity) depends on the conditions of the electrodialysis process and the impact of various degradation products of amines on them. In this regard, the purpose of this work was an experimental study of the process of thermal degradation of MEA solutions and the effect of the products of this process on the separating properties of ion-exchange membranes (performance and selectivity) on the efficiency of their electrodialysis purification. The aim of this study was to assess the prospects for the electrodialysis process from the point of view of the performance of cation- and anion-exchange membranes under conditions of prolonged contact with a working solution containing MEA thermal-oxidative degradation products. Moreover, in this work, for the first time, an approach to the regeneration of ion-exchange membranes during electrodialysis was experimentally tested.

## 2. Materials and Methods

### 2.1. Materials

For the preparation of model absorbent solution, the following chemicals were used: monoethanolamine (chemical grade, Chimmed, Podolsk, Russia), carbone dioxide (technical grade, MGPZ, Moscow, Russia), formic acid (chemical grade, Khimmed Synthes, Podolsk, Russia), gladical acetic acid (chemical grade, Khimmed Synthes, Podolsk, Russia), oxalic acid dehydrated (chemical grade, Khimmed Synthes, Podolsk, Russia), nitric acid (chemical grade, Khimmed Synthes, Podolsk, Russia), sulfuric acid (chemical grade, Khimmed Synthes, Podolsk, Russia), distilled water. A 30% wt. solution of MEA in water was prepared gravimetrically; then, carbon dioxide was introduced to achieve the required CO_2_-loading.

Ion-exchange membranes MA-41 and MK-40 (Schekinoazot, Schekino, Russia) have the characteristics presented in Table 1. Before being used in experiments on electrodialysis purification of carbonized MEA solutions, ion-exchange membranes were kept in distilled water for 24 h.

### 2.2. Thermal Exposition of MEA Solution

The aging of the carbonized MEA solutions was carried out using a sealed stainless steel vessel (5 L). The vessel, half-filled with a solution of carbonized MEA, was placed in a thermostat, and exposure was carried out for 336 h (two weeks) at 120 °C. Sampling from a vessel (preliminarily cooled to room temperature) to assess the degree of degradation of the MEA solution was carried out after 7, 13, 45, 75, 155, 215, and 336 h using the spectrophotometer PE-5400UF (Promecolab, St. Petersburg, Russia). The measurements were carried out in a wide range of wavelengths: from 200 to 1000 nm. After filtering the solution through a UFFK-1 microfilter (average pore size 240 nm, Vladipor, Vladimir, Russia), the color was eliminated, and the sediment content was 20 g/L.

To ensure the identity of the resulting aged solution in real conditions of the carbonization process (and it takes place in the presence of oxygen contained in the flue gas), before experiments with ion-exchange membranes, MEA oxidation products were added to the filtered solution—thermostable salts (TSS) of such acids as, mg/L: formic 800, oxalic 330, acetic 70, nitrogen 460, sulfuric 670, and hydrochloric 70.

### 2.3. Exposure of Ion-Exchange Membranes in Degraded MEA Solution

First of all, ion-exchange membranes were kept in distilled water for 24 h. Then, the membranes were placed in a degraded MEA solution (336 h) and kept in continuous contact with it for 6 months (4392 h). The choice of a solution corresponding to 336 h of degradation is determined by the degree of its degradation (the presence of a significant amount of sediment, the presence of metals in the solution, and MEA compaction products). Such a solution makes it possible to clearly demonstrate the effect of soluble degradation products on the transfer of ions through membranes during electrodialysis purification.

### 2.4. Electrodialysis

To assess the electrokinetic activity of MEA degradation products, a comparison was made of the electrodialysis treatment of a model 30% wt. carbonized aqueous MEA solution with a residual carbon dioxide content of 0.2 mol CO_2_/mol MEA using fresh MK-40 and MA-41 membranes and the same membranes after 6 months of contact with a pre-degraded MEA solution. The flow scheme of the electrodialysis lab set-up is represented in Figure 1. The flat-type electrodialyzer with an active surface area of about 20 dm^2^ was equipped with commercial cation-exchange membranes MK-40 and anion-exchange membranes MA-41 (ShchekinoAzot Ltd., Pervomayskiy, Russia). The alternation of membranes led to the formation of 10 circulated desalting cells and 9 flowing concentrating cells. Both electrodes were made of platinized titanium. The rinse of electrode compartments was 30% wt. of MEA in water. Model MEA solution was poured into container V1 in the amount of 1 l and into the containers of anolyte (V2) and catholyte (V3). The concentrate tank (V4) was filled with distilled water, which made it possible to create the maximum difference in the electrical conductivity of the solution in the desalination and concentration chambers. This technique led to the intensification of the process of electrodialysis isolation of the target components. Pumps P1-P4 provided continuous circulation of liquid in the hydraulic paths at a given speed. The average linear velocity of solutions in the sections of the electrodialyzer was 0.35 cm/s (Re ≈ 10). Electrodialysis was carried out in the controlled potential mode at 30 V voltage. The DC Power Supply HY5005E-2 (Mastech, San Jose, CA, USA) power supply unit supplies the electrodialyzer with direct current. The current flowing through the machine was detected by a DC source. The intensity of the electrodialysis process was estimated from the change in the electrical conductivity of the diluate solution and its temperature (fixed according to the readings of the multifunctional device C every 10 min of the experiment).

The specific energy consumption (*Qs*, kJ/g(HSS)) for the ED process was calculated based on experimentally obtained data according to Formula (1):(1)$$Qs=\frac{U\cdot I\cdot t}{m(HSS)}$$
where *U* (V) is the voltage applied to the electrodes, *I* (A) is the average current over a period of time *t* (h) and m (*HSS*) (g) is the *HSS* mass removed over a period of time *t* (h).

The depth of desalination (release of carbon dioxide) was estimated from the values of the specific electrical conductivity (SEC) of solutions and the dependence of the SEC of a carbonized MEA solution on the degree of saturation with carbon dioxide [15]. Since in the studied range of carbonization degree (0.25–0.1 mol CO_2_/mol MEA) the dependence of electrical conductivity on CO_2_ concentration is linear, the calculation of the desalination depth can be carried out using the electrical conductivity values according to Formula (2):(2)$$\Delta C=\frac{{SEC}_{0}-{SEC}_{60}}{{SEC}_{0}}\times 100\%$$
where η is the depth of desalination; *SEC*—electrical conductivity; 0, 60 min is the duration of the electrodialysis experiment.

The current efficiency of carbonate ions η in the electrodialysis apparatus was calculated using Formula (3):(3)$$\eta =\frac{Fn}{{IN}_{cell}\tau }$$
where *I* is the current supplied to the apparatus, A; *N_cell_* is the number of elementary cells in the membrane package of the electrodialysis apparatus; *n* is the amount of obtained carbonate ions, mol-eq; *τ* is the time elapsed since the beginning of the experiment, s; *F* is Faraday’s constant, 96,485 A s/mol.

### 2.5. X-ray Fluorescence Analysis

The elemental composition of the precipitate in degraded MEA was analyzed using wavelength-dispersive X-ray fluorescence analysis, carried out on an ARL PERFORM’X (Thermo Fischer Scientific, Waltham, MA, USA) sequential X-ray fluorescence spectrometer using a rhodium tube [35]. Up to 79 elements were analyzed and the percentage composition of the sample was calculated using the UniQuant program to a relative error of 5%.

## 3. Results and Discussion

To study the electrokinetic activity of degradation products, this work was divided into two parts: (1) obtaining a degraded amine solution; (2) comparison of the efficiency of electrodialysis treatment of a carbonized amine solution before and after contact of ion-exchange membranes with a degraded solution. The article also considers the approach of regeneration of ion-exchange membranes from degradation products.

### 3.1. Thermal Aging of MEA Solutions

The appearance of a rich brown color and turbidity of the MEA solution during its thermal exposure at a temperature of 120 °C is direct evidence of the thermal aging processes of MEA. The detection of iron and nickel cations in the filtered sediment (Table 2) is obviously a consequence of corrosion of the stainless steel from which the vessel is made (we note in brackets that the final concentration of insoluble impurities in the aged solution was 20 g/L).

The graphical dependence of the increase in the concentration of insoluble iron and nickel compounds, built on the basis of the data in Table 1, indicates the presence of an incubation period (0–75) h, after which the concentration of cations of these metals increases exponentially (Figure 2).

A significant excess of the concentration of iron in the oxidation products is understandable since nickel is initially an additive in the composition of stainless steel. It should be noted that the nature of the dependences obtained correlates with the data [9] obtained during the testing of a pilot plant. A comparative analysis of the above dependences shows that the nature of the curves for Fe, Cr, and Ni is the same. The higher absolute values of the concentrations of Fe and Ni cations are explained by the fact that in [9], the results of the absorption of CO_2_ from a real flue gas in which oxygen is present are presented, while in the experiments of this work, model solutions of carbonized MEA were used, which were not saturated with oxygen. According to the data given in [5], iron ions are catalysts for the oxidative degradation of MEA. Therefore, this circumstance leads to an exponential increase in the concentration of both iron and HSS anions (formates, oxalates et al.).

Analysis of the change in the optical density of the MEA solution during degradation (Figure 3) demonstrates that thermal degradation processes are already observed after the first 13 h of exposure.

The data in Figure 3 indicate that MEA degradation begins almost in the first h (13 h) of thermal exposure to the solution. An analysis of changes in the composition of the solution before (curve 336 h) and after (curve 336 h.f.) the removal (filtration) of suspended particles indicates that the precipitate appears in the range of 75–155 h of thermal exposure of the solution since the curves of changes in the optical density of solutions (155 h and 336 h.f.) are almost identical. Joint data analysis in Figure 2 and Figure 3 and data from the literature show that a sharp jump in the concentration of iron and nickel ions is observed in the area after 75 h. These data correlate with the data given in [3,5] and allow us to formulate a practical recommendation for the process of electrodialysis purification of the entire volume of the MEA solution—it should be started at a time interval no later than 50 h and preferably at the same time as the start of the absorption process.

### 3.2. Electrokinetic Activity of MEA Degradation Products during Electrodialysis

The course of the electrodialysis process was assessed by the change in the electrical conductivity of the diluate solution and its temperature, which were recorded according to the readings of the conductometer (“C”, Figure 1) every 10 min of the experiment. Figure 4 presents comparative data on the operation of a laboratory electrodialyzer for fresh ion-exchange membranes and for membranes after a 6-month exposition in degraded MEA. The dependence of the electrical conductivity of the treated solution on the time of electrodialysis clearly demonstrates the decrease in transfer through the membranes.

Desalting depth for MA-41 and MK-40 membranes after exposure to the degraded MEA solution decreased to 1.6 mS/cm compared to 4.4 mS/cm for fresh membranes (Figure 4). It should be noted that the current flowing through the membrane also decreased from 1.9 A for fresh membranes to 1.6 A for membranes after contact with the degraded MEA solution (Figure 5).

This decrease in productivity is due to clogging of the membrane with amine degradation products. It should be noted that the deposition of degradation and corrosion products on the surface of ion-exchange membranes leads to a change in their color (Figure 6).

It can be seen that corrosion and MEA degradation products are deposited on the membrane surface. Obviously, the amine compaction products adsorbed on the membrane surface have low electrokinetic activity (Figure 4 and Figure 5). This has a negative effect on the efficiency of the membranes: the current efficiency, the magnitude of the current strength, and the degree of desalination of the solution decrease in comparison with the original membranes. As a result, it is important to regenerate ion-exchange membranes after fouling.

### 3.3. Regeneration of Ion-Exchange Membranes

In this work, for the first time for ion-exchange membranes MK-40 and MA-41, alkaline regeneration was carried out from corrosion products and degradation of the MEA absorbent during electrodialysis. Since the products of MEA compaction in aqueous solutions are cationic in nature, they preferentially settle on cation-exchange membranes during electrodialysis. The sediment accumulation on the ion-exchange membrane leads to an increase in its resistance and, as a result, to a decrease in the module amperage (which we observed on membranes that were in contact with the degraded solution for 6 months). In order to assess the contribution of the effect of alkanolamine compaction products on the operation of ion-exchange membranes, a 4-fold regeneration of membranes with an alkali solution (with a concentration of 5 g/L NaOH) was carried out for the first time in the electrodialysis mode. The choice of such a regenerating agent is due to the fact that cation-exchange membranes are supplied in the Na-form, that is, the counterion of the polyelectrolyte in the membrane is Na^+^. In this case, a highly alkaline environment contributes to the washing off of deposits from the membrane surface. Thus, one can expect a decrease in the resistance of the membrane and an increase in its conductivity.

Electrodialysis of an alkaline solution of 5 g/L NaOH was carried out for 1 h. An aqueous solution with an alkali content of 5 g/L NaOH was supplied to the desalination chambers, and distilled water was supplied to the concentration chambers. At each repetition, the hydraulic circuits for desalting and concentrating were filled with fresh alkali solution and distilled water. The electrode chambers were filled with a sodium sulfate solution, which ensured high conductivity in the electrode chambers. The kinetics of changes in electrical conductivity for membranes before and after regeneration is shown in Figure 7.

It can be seen that after 6 months of exposure, the resistance of ion-exchange membranes increases. Clogging of the membrane surface with groups with low electrokinetic activity leads to a decrease in the transfer of carbonate ions through the membranes. This behavior persists during repeated electrodialysis treatments of the MEA absorbent on clogged membranes. A single regeneration of the membranes with a NaOH solution made it possible to reduce the gap between fresh and clogged membranes in terms of the depth of desalination. However, 4-fold electrodialysis treatment of an alkaline solution (5 g NaOH/L) allows for increasing the transfer through the membranes (Figure 7). Apparently, as a result of regeneration, the resistance of the membranes decreases. Such a conclusion can be drawn by analyzing the change in current strength and current output during electrodialysis (Figure 8).

As noted earlier, after exposure of ion-exchange membranes in a model absorption solution of MEA, the depth of desalination decreased by 34%, while the magnitude of the current flowing through the apparatus decreased by 25%. A single regeneration with an alkaline solution made it possible (Figure 7 and Figure 8) to reduce this gap to 14.8. However, no noticeable change in current strength was observed. An increase in the number of regeneration cycles up to 4 made it possible to reduce the decrease in the CO_2_ extraction depth to 10%. It should be noted that the current after 4-fold alkaline regeneration reached 1.5 A; however, the transfer of carbonate ions remained higher than in the case of clogged membranes. This is also confirmed by the dependence of the current output on the time of electrodialysis (Figure 8b). The dependence of the flow rate after 1 regeneration cycle was higher in the case of clogged membranes. Four successive regeneration cycles made it possible to multiply the current output at the initial stage of electrodialysis. However, by the end of the hourly cycle, the current efficiency decreased, but remained higher than in the case of a single regeneration. A schematic representation of the expected transfer of ions in the process of alkaline membrane regeneration is shown in Figure 9. A comparison of the depth of CO_2_ removal, as well as the integral current efficiency during the electrodialysis of the model MEA solution on the initial, clogged, and regenerated membranes, is presented in Table 3.

When analyzing the performance of ion-exchange membranes per hour of electrodialysis, it becomes obvious that the regeneration performed makes it possible to increase their efficiency. It should be noted that alkaline regeneration made it possible to achieve a high degree of restoration of the ion-exchange properties of membranes (relative extraction depth 90%, current efficient 96%). This allows us to state that the clogging of ion-exchange membranes occurs mainly by products of thermal degradation of MEA.

## 4. Conclusions

A study of the thermal degradation of a carbonized 30% wt. solution of monoethanolamine (MEA) for 336 h at 120 °C was made. Based on the change in color of the solution and the formation of a precipitate, the thermal degradation of the MEA solution was confirmed. As part of the study of the thermal degradation of an MEA solution, the electrokinetic activity of the tarry products of densification of the amine in the degraded solution was studied. To determine the electrokinetic activity of the amine compaction products, a package of MK-40 and MA-41 ion-exchange membranes was exposed to a degraded solution for 6 months. A comparison of the electrodialysis treatment of a model MEA absorption solution before and after exposure showed that the relative depth of desalination is reduced by 34%, while the magnitude of the current flowing through the apparatus is reduced by 25%. For the first time, the alkaline regeneration of ion-exchange membranes was carried out in the mode of electrodialysis from MEA degradation products, which made it possible to restore the relative depth of desalting to 90% and relative current efficiency to 96%. It has been shown that to optimize the process of electrodialysis regeneration of the MEA solution, it should be started simultaneously with the start of the carbon dioxide absorption process.

## Data Availability

The data presented in this study are available on request from the corresponding author.

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
