# Peer review of "Monoethanolamine (MEA) Degradation: Influence on the Electrodialysis Treatment of MEA-Absorbent"

_membranes, 2023, doi:10.3390/membranes13050491_

Round 1
Reviewer 1 Report
In this paper, the authors report the effect of the thermal oxidation decomposition of monoethanolamine (MEA) on the performance of ion-exchange membranes and the regeneration of membranes through electrodialysis. The content of this paper is new and interesting but it needs to be revised in several parts for publication in membranes. My detailed comments are as follows:
- (Table 1) Please, check the unit of surface electrical resistance. Ohm/cm2 or Ohm cm2.
- A schematic draw showing the transport of ions that can show the principle of electrodialysis treatment (for regeneration) should be added.
- hours -> h, minutes -> min
- (L 284, 289) fig. 3 -> Figure 3, fig. 2 and 3 -> Figures 2 and 3
Author used the terms "Figure", "fig.", and "Fig." interchangeably, but I recommend unifying them with "Figure".
- It is necessary to compare the transport numbers and I-V curves before and after the exposure of membranes to a degraded solution and after the regeneration.
Author Response
Thank you for your review and your time for our article. We respond to your comments below.
- It is necessary to compare the transport numbers and I-V curves before and after the exposure of membranes to a degraded solution and after the regeneration.
Thank you very much for your kind report.
- (Table 1) Please, check the unit of surface electrical resistance. Ohm/cm2 or Ohm cm2.
Thanks for the comments. The correction has been made to the text of the article.
- A schematic draw showing the transport of ions that can show the principle of electrodialysis treatment (for regeneration) should be added.
A figure has been added to the text of the article illustrating our understanding of the regeneration process – Figure 9.
- hours -> h, minutes -> min
Thanks for the comments. The correction has been made to the text of the article.
- (L 284, 289) fig. 3 -> Figure 3, fig. 2 and 3 -> Figures 2 and 3
Author used the terms "Figure", "fig.", and "Fig." interchangeably, but I recommend unifying them with "Figure".
Thanks for the comments. The correction has been made to the text of the article.
- It is necessary to compare the transport numbers and I-V curves before and after the exposure of membranes to a degraded solution and after the regeneration.
Thank you for your comment. Taking into account the selected device configuration (close to industrial ED devices), direct measurement of the transport numbers and I-V curves is not possible. The resulting CVC character is not informative (the figure is shown below). Since the work is of a technological nature, we studied the effect of degraded MEA deposits on the properties of the entire apparatus. The study of transfer numbers in an elementary electrochemical cell was not among the current tasks, since it is more related to the study of the properties of individual membranes. This is indeed of interest, we will consider this issue in the course of further work.
Figure. I-V curve for ED module with initial membranes (blue) and with membranes after contact degraded MEA (dMEA) (red).

Reviewer 2 Report
What are the criteria for determining the times 7, 13, 45, 75, 155, 215, and 336 hours? There seems to be no rules in the selection of these times.
Author Response
Thank you very much for your kind report.
This time was chosen based on the length of the working day and the possibility of sampling and characterization during working hours. Since an insoluble precipitate was observed at 336 hours, the amine degradation experiment was terminated. Precipitation was an indication of a high degree of degradation of the solution.
Reviewer 3 Report
This manuscript investigated the thermal degradation process of MEA solutions and the effect of the products of this process on the separating properties of ion-exchange membranes on the efficiency of their electrodialysis purification. The experimental approach is systematic and contents are organized, and thus I think the manuscript could be accepted in Membrane.
Specific question:
Figure.8 Why the trend of the current efficiency of the regeneration 4 cycles behaved different with other samples? Could you please give an explanation?
Author Response
Thank you for your kind review of our work. Regarding your question: the trend divergence is observed in the first 10 minutes of operation of the electrodialysis machine. At this time, the device enters the mode and such a discrepancy is permissible, taking into account the previous alkaline treatment of the membranes.
Round 2
Reviewer 1 Report
The authors have carefully revised the manuscript according to the referees’ comments. In my opinion, this manuscript could be accepted for publication in membranes.